# Close to the border—Resilience in healthcare in a European border region: Findings of a needs analysis

**Leonie A. K. Loeffler**[1,2], **Sophie Isabelle Lambert**[1,2], **Lea Bouché**[1,3], **Martin Klasen**[1,2], **Saša Sopka**[1,2], **Lina Vogt**[1,2]*, **COMPAS Consortium**[¶]

1 Department of Anesthesiology, University Hospital RWTH Aachen, Medical Faculty, RWTH Aachen University, Aachen, Germany, 2 AIXTRA–Competence Center for Training and Patient Safety, Medical Faculty, RWTH Aachen University, Aachen, Germany, 3 ARS–Aachen Institute for Rescue Management and Public Safety, City of Aachen and University Hospital RWTH Aachen, Aachen, Germany

¶ The complete members of the COMPAS consortium can be found in the Acknowledgments.
* lvogt@ukaachen.de

**Data Availability Statement:** All data files are uploaded as supplementary materials.

**Funding:** This study was financed by the Interreg V-A Euregio Meuse-Rhine (EMR) programme

## Abstract

### Objectives

Promoting resilience, the ability to withstand and overcome challenging situations, is crucial for maintaining the performance of healthcare systems. Unique challenges faced by healthcare facilities in border regions render them particularly vulnerable during crises, emphasizing the need to promote resilience in these areas. The current study evaluated the state and needs of resilience in healthcare professionals in a representative European border region.

### Methods

All hospitals and emergency medical care services in the Euregio Meuse-Rhine (Germany, Belgium, the Netherlands) were approached to participate via an online-survey. Behavioral data on psychological distress (Patient Health Questionnaire-4), work-related stressors, individual resilience (Brief Resilience Scale, Resilience at Work scale), and organizational resilience (Benchmark Resilience Tool-short) were collected.

### Results

2233 participants initiated the survey with 500 responses included in the analysis. 46% of the participants indicated clinically significant psychological distress. Most challenging stressors were staff availability, available time, and workload. On average, individual resilience was in the normal range, yet 15.6% showed below average resilience. At the organizational level, healthcare institutions can particularly enhance resilience in the domains of *Internal resources*, *Situation Awareness*, and *Unity of purpose*. Compared to their neighbor countries, German healthcare professionals indicated higher levels of depressive symptoms, were more burdened by work-related stressors, and reported lower levels of organizational resilience.

([https://www.interregemr.eu/home-en](https://www.interregemr.eu/home-en); project number EMR208 to S.S.). The Interreg V-A Euregio Meuse-Rhine (EMR) programme invests almost EUR 100 million in the development of the Interreg-region from 2014 until the end of 2023. With the investment of EU funds in Interreg projects, the European Union directly invests in the economic development, innovation, territorial development and social inclusion and education of this region. The study was further co-financed by regional authorities in Germany (Ministry of Economic Affairs, Industry, Climate Action and Energy of the State of North Rhine-Westphalia), Belgium (Wallonia), and the Netherlands (Province of Limburg). The funders had no role in study design, data collection and analysis, decision to publish, or preparation of the manuscript.

**Competing interests:** The authors have declared that no competing interests exist.

## Conclusion

Findings highlight that healthcare institutions not only need to promote the resilience of the individual employee particularly in border regions, healthcare institutions, must also act to be better prepared for potential threats and crises while considering each country's unique needs.

## 1. Introduction

Healthcare professionals face numerous stressors in daily work life [1], either directly related to patient care, such as confrontation with illness and death, or structural stressors like lack of staff or time [1, 2]. Impaired health, absenteeism, and job drop-out may result and affect healthcare professionals' ability to deal with critical situations [3, 4]. Consequently, patients' and employees' safety may be at risk. Hall and colleagues, for instance, revealed a link between healthcare professionals' burnout and poor patient safety outcomes, including medical errors [4].

Resilience protects against harmful impacts of (occupational) stress and enables sustained performance during crises [5]. In healthcare, resilience refers to the ability or capacity to withstand and overcome challenges and threats safeguarding patient care and employee wellbeing [6]. Resilience operates on three levels: On the individual level, resilience can be enhanced by developing personal resources and resilient behaviors (e.g., adaptive coping strategies) [6]. On the team level, resilience can be fostered by targeting team/intergroup climate, dynamics, and processes (e.g., psychological safety, connectivity, debriefings) [6]. Finally, resilience can be nurtured on the organizational level, through systematic efforts to strengthen individual and collective resources (e.g., financial, structural, human, social) and the establishment of proactive practices (e.g., strategic planning) [6]. In a review, Wallace and colleagues [1] revealed a beneficial impact of interventions promoting individual resilience like stress management on healthcare quality. Likewise, healthcare professionals facing impending crises indicated (de) briefings, clear and simple protocols, continuously adapted procedures, as well as support by colleagues for reducing stress and strengthening overall effectiveness [7].

To enhance resilience of healthcare professionals and healthcare institutions, potential targets for improvement must be identified. These targets might vary depending on regional and national contexts, emphasizing the necessity to consider local needs. Border regions, compared to inland areas, exhibit distinct characteristics. In economics, the special needs and characteristics of border regions have been subject. This debate has two approaches: firstly borders lead to regions trading disproportionately with other regions in the same country and not with those that are the same distance away but on the other side of the border [8]. Secondly border regions tend to be less endowed with growth-promoting factors [9] or utilize similar endowments less efficiently [9]. Economic growth, in turn, appears to be linked to improvements in the healthcare system, for example due to better public healthcare infrastructure [10]. Hence, research about patient care in border regions is important, to identify factors for improvements to enhance efficiency and quality of patient care.

Looking back at the COVID 19 pandemic border regions are particularly vulnerable during crises [11]. They not only tend to be economically weaker and have less developed infrastructure [12, 13], but effective crisis management in such regions necessitates coordinated efforts between involved countries, a deep understanding of administrative and economic conditions on both sides of the border, and the ability to overcome intercultural barriers due to language,

administrative procedures, habits and standards [11]. Recent research highlights disparities in resilience between border and non-border regions within the European Union (EU), with border regions exhibiting lower resilience and heightened susceptibility to crises in the short-term [14]. Consequently, these regions stand to benefit significantly from interventions aimed at bolstering resilience. Particularly in healthcare within border regions, it is crucial to prioritize resilience-building efforts to ensure continued and safe provision of patient care in times of crises [15].

Europe with its 47 countries is marked by 45 land border regions and 17 maritime border regions, meaning 62 overall border regions. Within this region a relatively high population density exists, presenting unique challenges for healthcare institutions. In more detail, cross-border patient care and communication impose additional burdens on healthcare staff due to higher administrative costs arising through missing documents, language barriers, cultural disparities, and heightened demands on hospital infrastructure [16, 17]. These challenges are further compounded during crises. For instance, border closures implemented during the COVID-19 pandemic impeded cross-border collaboration and personnel mobility, thereby intensifying the burden put on healthcare professionals [12]. Asymmetric decisions made by authorities of the different countries further exacerbated the administrative load [12].

Due to the multitude of unique challenges faced by healthcare professionals and facilities in border regions, it is crucial to comprehend their needs and prioritize efforts to promote resilience. This study examined the current state and needs regarding resilience in healthcare in the three-country European border region Meuse-Rhine ("Euregio Meuse-Rhine", EMR; Belgium, Germany, and The Netherlands). Specifically, it investigated psychological distress, work-related stressors, and both individual and organizational resilience among healthcare professionals and institutions, while also addressing country-specific needs. Findings do not only serve healthcare facilities in the EMR as basis for resilience-promoting measures but also provide valuable insights to other European border regions facing similar border-region specific challenges (e.g., Rhine-Meuse-North Euregio; see [18] for details of border regions in Europe).

## 2. Materials and methods

### 2.1. Sample

A total of 49 hospitals and 21 emergency medical services in the EMR were approached for participation. The EMR spans from Leuven (Belgium) in the West to the borders of Cologne (Germany) in the East and from Eindhoven (The Netherlands) in the North to the border of Luxemburg in the South. The EMR is home to over 5.5 million people in three countries, comprising both urban agglomerations and rural areas [18]. Therefore, the study findings are likely representative of urban border regions in Europe [18]. Heads of the respective facilities were contacted with the request to forward an online survey to all employed healthcare professionals. Data collection took place from July 25th to November 25th, 2022.

### 2.2. Sample size planning

Sample size planning was not performed as we intended to include the entire target population (i.e., all hospitals and emergency medical services in the EMR). However, assuming a medium effect size (i.e., $d = .5$ [Mann-Whitney-U test], $f = .25$ [Kruskal-Wallis test]), a significance level of $\alpha = .05$, and a power $(1-\beta)$ of 95%, a minimum number of 220 participants was deemed necessary for Mann-Whitney-U tests and 252 participants for Kruskal-Wallis tests using G*Power 3.1.

## 2.3. Ethical approval

This study was conducted in accordance with the Declaration of Helsinki and approved by the Institutional Review Boards of the University Hospital RWTH Aachen, Germany (EK502-21), the University Hospital Liège, Belgium (2022/11), and the Maastricht University, The Netherlands (METC 2022–3108). Participants were informed about the study on the starting page of the online survey and consented to participate in the study by starting the survey. Data was collected anonymously.

## 2.4. Measures

The online survey consisted of standardized and validated questionnaires along with newly developed questions, provided via SoSci Survey (Version 3.1.06). Questions without official translations were translated into German, English, French, and Dutch and cross-checked by native speakers. Dutch institutions received a shortened questionnaire with fewer resilience questions to address concerns of participant overburdening.

**2.4.1. Psychological distress.** Psychological distress was examined using the Patient Health Questionnaire 4-scale (PHQ-4, 4 items) [19]. The PHQ-4 includes the Generalized Anxiety Disorder 2-scale (GAD-2, 2 items) and the Patient Health Questionnaire 2-scale (PHQ-2, 2 items). The GAD-2 and the PHQ-2 provide valuable ultra-brief screenings for anxiety and depression, respectively, while the PHQ-4 total score provides an overall measure of clinically relevant symptom burden (i.e., psychological distress). Participants rated the frequency of specific symptoms on a scale from 0 to 3 (not at all—almost every day).

**2.4.2. Work-related stressors.** Work-related stressors were assessed by showing participants a list of common stressors in healthcare (15 items), like *Workload*, *Interactions with patients and relatives* or *Physical exertion* [2]). Participants rated the extent to which they felt burdened by these stressors on a scale from 1 to 6 (strongly disagree—strongly agree).

**2.4.3. Individual resilience.** Individual resilience was examined using the Brief Resilience Scale (BRS, 6 items) [20] and the Resilience at Work Scale (RAW scale, 20 items) [21]. The BRS measures the general ability of an individual to bounce back or recover from stress. Participants indicated their agreement with resilience statements on a scale from 1 to 5 (strongly disagree—strongly agree). The RAW scale assesses an individual's work-related resilience. Participants indicated their agreement with resources linked to resilience on a scale from 0 to 6 (strongly disagree—strongly agree).

**2.4.4. Organizational resilience.** Organizational resilience was assessed using a short version of the Benchmark Resilience Tool (BRT-short, 13 items) [22–24]. Participants indicated their agreement with 13 organizational resilience indicators on a scale from 1 to 8 (strongly disagree—strongly agree). These indicators are divided under three categories:

1. "Leadership and Culture", i.e. adaptive capacity of an organization created by its leadership and culture,

2. "Network", i.e., internal/external relationships fostered and developed by an organization which can leverage when needed, or

3. "Change Ready", i.e., planning and alignment to enable organizational readiness for change [22].

## 2.5. Statistical analysis

Quantitative data were analyzed using IBM SPSS Statistics Version 28 (IBM Corp., Armonk, NY, USA) applying an $\alpha$-level of $p < .05$. Sum scores for psychological distress (PHQ-4) and mean scores for stressors and resilience (BRS, RAW scale, BRT-short) were calculated.

Participants with missing values in sum score calculations (e.g., PHQ-4) were excluded from further analysis of the respective questionnaire. Since the data were not normally distributed, country-specific effects were examined using Mann-Whitney-U tests (i.e., PHQ-4, BRS, RAW scale, BRT-short) and Kruskal-Wallis tests (i.e., stressors). Country served as independent factor and psychological distress (PHQ-4), stressors at work, and resilience (BRS, RAW scale, BRT-short) as dependent factors. Dunn's Post Hoc tests were computed for identifying significant country differences (i.e., stressors). Moreover, Spearman correlations assessed the relationship between psychological distress (PHQ-4) and resilience parameters (BRS, RAW scale, BRT-short).

## 3. Results

### 3.1. Sample characteristics

A total of 2233 healthcare professionals initiated the survey (Fig 1). Data of $n = 500$ healthcare professionals (Mean age = 44.16 years, 62.1% female / 35.3% male / 0% non-binary identity / 2.6% do not want to indicate were analyzed. Details on the distribution of the sample across healthcare professions and countries are presented in Fig 2.

### 3.2. Psychological distress (PHQ-4, n = 466)

Nearly half (46%) reported clinically relevant psychological distress (PHQ-4 $\geq$ 3). Moreover, 22.1% of participants screened positive for an anxiety disorder (GAD-2 $\geq$ 3) and 15.9% for a depressive disorder (PHQ-2 $\geq$ 3; Table 1).

### 3.3. Work-related stressors (n = 490)

Participants indicated to be most burdened by personnel availability, available time, and workload (Table 2).

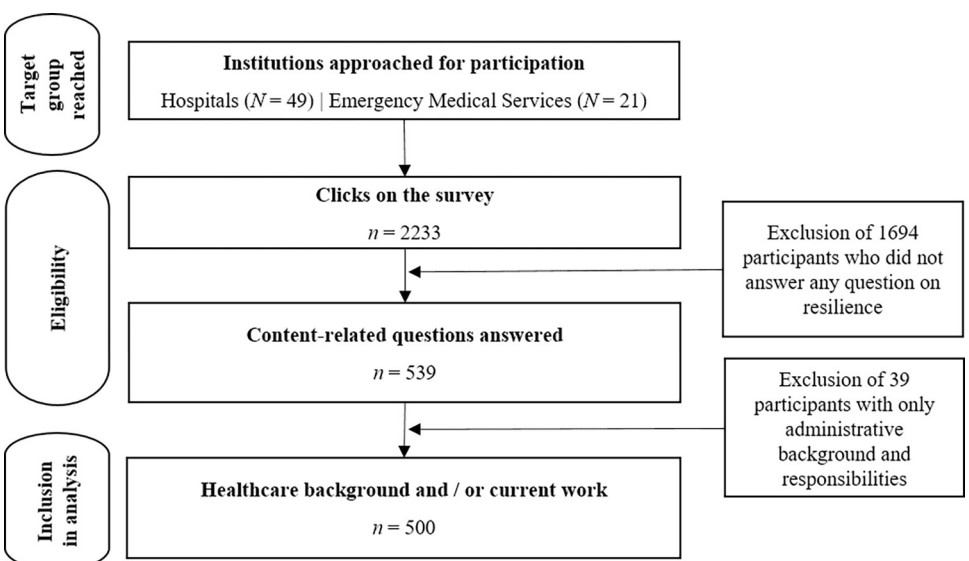

**Fig 1. Flow chart of included participants.** 2233 participants started the survey. 1694 participants were excluded as they did not provide valid answers or did not answer any question on resilience. 39 participants were excluded as they solely had an administrative background and responsibilities (i.e., neither medical training nor medical activity). In total, data of 500 participants were included in the analyses.

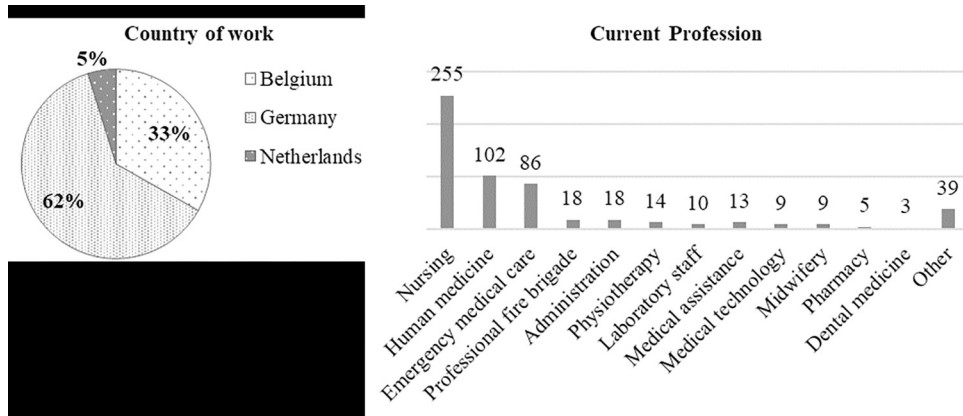

**Fig 2. Sample characteristics.** Country of work (in percentage; left graph) and current profession (number of participants; right graph) of included participants (*n* = 500). Participants could indicate multiple answers regarding the profession.

## 3.4. Individual resilience

General individual resilience (BRS, *n* = 468): On average, participants reported normal individual resilience (Mean = 3.55, *SD* = .68). 15.8% had low individual resilience (BRS ≤ 2.99). 71.6% reported average individual resilience scores (BRW = 3.00–4.30), and 12.6% exhibited high individual resilience (BRS ≥ 4.31).

Work-related individual resilience (RAW scale, *n* = 475): Compared to preliminary normative data, participants reported average work-related individual resilience (Mean = 3.99, *SD* = .68; Standardized Mean = 66.37, *SD* = 11.19).

## 3.5. Correlations psychological distress and individual resilience

General psychological distress (PHQ-4) correlated negatively with general individual resilience (BRS) ($r$ = -.473, $p \leq$ .001) and work-related individual resilience (RAW scale) ($r$ = -.464, $p \leq$ .001). Participants with high individual resilience reported less psychological distress.

**Table 1. Psychological distress.**

| | *n* | % | Country differences[1] |
|---|---|---|---|
| **Psychological distress (PHQ-4)** | | | *ns* |
| No | 251 | 54.1 | |
| Mild | 139 | 30.0 | |
| Moderate | 52 | 11.2 | |
| Severe | 22 | 4.7 | |
| **Anxiety (GAD-2)** | | | *ns* |
| No | 363 | 77.9 | |
| Yes | 103 | 22.1 | |
| **Depression (PHQ-2)** | | | DE > BE*** |
| No | 392 | 84.1 | |
| Yes | 74 | 15.9 | |

[1] Data only available for Germany (DE) and Belgium (BE); *n*, number of participants; *ns*, no significant difference between countries

* $p <$ .05, ** $p <$ .01, *** $p <$ .001

Note: country differences are calculated using sum scores of the PHQ-4, GAD-2, and PHQ-2.

**Table 2. Work-related stressors.**

| Stressor | Mean (*SD*) | Median | Country differences[1] |
|---|---|---|---|
| Personnel availability | 4.58 (1.43) | 5 | DE > NL / BE*** |
| Available time | 4.37 (1.44) | 5 | *ns* |
| Workload | 4.32 (1.47) | 5 | *ns* |
| Administrative expense | 3.87 (1.73) | 4 | DE / NL > BE*** |
| Financial reward | 3.61 (1.67) | 4 | DE > BE*** |
| Social expectations placed on healthcare professionals | 3.60 (1.65) | 4 | [DE > BE**] |
| Appreciation | 3.53 (1.61) | 4 | DE > BE / NL*** |
| Hierarchies | 3.22 (1.73) | 3 | *ns* |
| Expectations of patients | 3.16 (1.44) | 3 | *ns* |
| Responsibility for medical decisions | 3.13 (1.55) | 3 | *ns* |
| Physical exertion | 3.11 (1.61) | 3 | DE > BE*** |
| Patient illness, suffering, death and/or suicide of patients | 2.95 (1.42) | 3 | *ns* |
| Teamwork | 2.49 (1.40) | 2 | *ns* |
| Interactions with patients and relatives | 2.44 (1.38) | 2 | *ns* |
| Technological changes (e.g., new diagnostic tools) | 2.44 (1.40) | 2 | *ns* |

[1] Data available for Germany (DE), Belgium (BE), and the Netherlands (NL)

Question: "I feel burdened by the following work-related stress factors" (1 = strongly disagree– 6 = strongly agree)

*n*, number of participants who answered the question; *SD*, standard deviation; ns, no significant difference between countries

\* $p < .05$, \*\* $p < .01$, \*\*\* $p < .001$ (p-values for country differences refers to findings of Kruskal-Wallis test)

Note to country differences: After Bonferroni correction for multiple comparisons, country differences in "social expectations placed on healthcare professionals" are no longer significant and are therefore indicated in brackets.

Likewise, psychological distress (PHQ-4) correlated negatively with organizational resilience (BRT short) ($r$ = -.302, $p \leq .001$) suggesting that employees in more resilient organizations reported less psychological distress.

### 3.6. Organizational resilience (BRT-short, n = 466)

With respect to organizational resilience, participants indicated *Internal resources*, *Situation awareness*, and *Unity of purpose* as least pronounced resilience indicators in their organization (Table 3).

### 3.7. Comparison between countries

In the following sections, we indicate in brackets the countries with available data.

**Psychological distress (PHQ-4; Germany, Belgium).** Healthcare professionals in Germany indicated significantly higher depression scores than those in Belgium (PHQ-2: $U$ = 18333.00, $Z$ = -4.832, $p \leq .001$). This country difference remained significant after Bonferroni correction for multiple comparisons. Overall psychological distress (PHQ-4: $p$ = .154) and anxiety scores (GAD-2: $p$ = .071) did not differ significantly between the two countries (Table 1).

**Work-related stressors (Germany, Belgium, The Netherlands).** With respect to work-related stressors, healthcare professionals in Germany and in the Netherlands were significantly more burdened by *Administrative expenses* than in Belgium ($H(2)$ = 94.558, $p \leq .001$). Furthermore, healthcare professionals in Germany were significantly more burdened by a lack of both *Personnel availability* ($H(2)$ = 45.647, $p \leq .001$) and *Appreciation* ($H(2)$ = 31.006, $p \leq .001$) than in the Netherlands and in Belgium. Finally, healthcare professionals in Germany

**Table 3. Organizational resilience (BRT-short).**

| Organizational resilience (BRT-short) | Mean (SD) | Median | Country differences[1] |
|---|---|---|---|
| Overall | 4.57 (1.56) | 4.61 | ns |
| Internal resources (Network) | 4.08 (1.97) | 4 | ns |
| Situation awareness (Leadership and Culture) | 4.12 (2.00) | 4 | BE > DE*** |
| Unity of purpose (Change Ready) | 4.24 (1.96) | 4 | ns |
| Stress testing plans (Change Ready) | 4.36 (2.20) | 5 | [BE > DE*] |
| Proactive posture (Change Ready) | 4.44 (1.96) | 5 | [BE > DE*] |
| Staff engagement (Leadership and Culture) | 4.50 (1.94) | 5 | ns |
| Planning strategies (Change Ready) | 4.52 (1.92) | 5 | [BE > DE*] |
| Leadership (Leadership and Culture) | 4.53 (2.25) | 5 | ns |
| Breaking silos (Network) | 4.54 (1.89) | 5 | ns |
| Leveraging knowledge (Network) | 4.88 (2.02) | 5 | ns |
| Innovation and creativity (Leadership and Culture) | 5.03 (1.91) | 5 | BE > DE*** |
| Decision making (Leadership and Culture) | 5.03 (2.06) | 5 | ns |
| Effective partnerships (Network) | 5.10 (1.93) | 6 | ns |

[1] Data only available for Germany (DE) and Belgium (BE).

Question: "To what extent do you agree or disagree with the following statements regarding your organization?" (1 = strongly disagree– 8 = strongly agree). Note: the higher the value, the more participants agree that the respective resilience indicator is pronounced in the organization they work in.

SD, standard deviation; ns, no significant difference between countries

* $p < .05$, ** $p < .01$, *** $p < .001$

Note: After Bonferroni correction for multiple comparisons, country differences in "proactive posture", "stress testing plans", and "planning strategies" are no longer significant and are therefore indicated in brackets.

were significantly more burdened by a lack of *Financial reward* ($H(2) = 27.763$, $p \leq .001$), *Physical exertion* ($U(2) = 23.664$, $p \leq .001$), and *Social expectations* ($H(2) = 10.295$, $p = .006$) than in Belgium. After Bonferroni correction for multiple comparisons, country differences for the stressor *Social expectations* did no longer reach significance. All other stressors did not differ between countries (all $p$'s $\geq .081$; see Table 2).

**Individual resilience (BRS, RAW scale; Germany, Belgium).** No significant differences between countries were found in general individual resilience (BRS: $p = .082$) and work-related individual resilience (RAW scale: $p = .435$).

**Organizational resilience (BRT-short; Germany, Belgium).** There was no significant difference in overall organizational resilience between countries (BRT short: $p = .136$). However, significant differences were found for the resilience indicators *Situation awareness* ($U = 19402.50$, $Z = -3.802$, $p \leq .001$), *Innovation and creativity* ($U = 19959.50$, $Z = -3.207$, $p = .001$), *Stress testing plans* ($U = 21563.00$, $Z = -2.111$, $p = .035$), *Proactive posture* ($U = 21590.00$, $Z = -2.094$, $p = .036$), and *Planning strategies* ($U = 21451.00$, $Z = -1.981$, $p = .048$). All five resilience indicators were less pronounced in Germany compared to Belgium. After Bonferroni correction for multiple comparisons, only country differences for *Situation awareness* and *Innovation and creativity* remained significant. All other resilience indicators did not differ significantly between countries (all $p$'s $\geq .214$; Table 3).

## 4. Discussion

This study aimed to investigate the current state and needs regarding resilience in healthcare in the European border region EMR. We examined psychological distress, work-related stressors, as well as individual and organizational resilience of healthcare professionals and institutions in this region while also exploring potential country-specific needs.

Almost half (46%) of the healthcare professionals reported psychological distress, with a significant proportion screening positive for anxiety (22.1%) or depressive disorders (15.9%). These numbers are slightly higher than those reported in original validation studies of the PHQ assessing psychological distress in primary care patients in the USA [25]. When compared to prevalence in the general German and Belgian population, screening results for anxiety and depressive disorders are considerably higher in our sample [26, 27]. Higher psychological distress in our study may be attributed to work-specific challenges of healthcare professionals. At the same time, the high burden during the late phase of the COVID-19 pandemic [28], especially when considering additional stress due to border-region challenges (e.g., border closures, [12, 16]), may account for the higher prevalence of mental impairment.

In a nation-wide survey conducted in the United Kingdom in 2022, the National Health Service revealed that more than half of healthcare professionals came "to work in the last three months despite not feeling well enough to perform their duties" [29]. Alarmingly, systematic reviews clearly link impaired wellbeing to reduced patient safety outcomes such as medical errors or hospital infections [4, 30]. High levels of psychological distress identified in our study highlight the need to strengthen mental health of healthcare professionals in border regions to ensure safe patient care. Interventions targeting healthcare professionals' wellbeing can indeed improve the quality of patient care [1]. Isaksson Rø and colleagues revealed a reduction of emotional exhaustion and sick leave at 1-year follow-up of a counselling intervention [31].

Major sources of occupational stress were identified, with a lack of personnel and time as well as a high workload being particularly burdensome. These stressors have been previously identified as obstructive [32] and seem to reflect the current shortage of qualified medical staff [33]. Such a shortage of qualified medical workers is exacerbated by high sick leave rates in the health and social care sector, leading to more vacant positions [34]. In addition, healthcare professionals in border regions such as the EMR are exposed to additional workload caused by cross-border patient care [16, 17]. Intriguingly, major stressors identified in our study point particularly towards organizational characteristics of healthcare institutions. Work directly related to patient care, such as interactions with patients and relatives, facing illness and suffering, or making medical decisions, were barely indicated as stressful. These findings emphasize the need for organizational changes in healthcare institutions to protect employee-wellbeing and ensure safe patient care [35]. In support, a systematic review and meta-analysis disclosed that interventions at the organizational level are more effective in reducing burnout among healthcare staff than individual-directed interventions [35].

At the organizational level, it is crucial for healthcare institutions, particularly in border regions, to enhance resilience in the face of crises. Our findings imply that healthcare institutions in the EMR need to establish *Situation awareness* by promoting vigilance and sharing of early warning signals among staff [22, 23]. Importantly, situational awareness should extend beyond individual healthcare institutions. A nationwide awareness of cross-border regional needs, which might be facilitated by mechanisms such as permanent contact points, is crucial [13]. Moreover, healthcare institutions should improve management and mobilization of resources for providing necessary capacity in the event of crisis (*Internal resources*) [22, 23]. Improving management and mobilization of resources might become particularly relevant for cross-border cooperation. Recommendations based on experiences of the COVID-19 pandemic include, improving information exchange regarding available cross-border resources [15], thereby potentially improving healthcare resilience in border regions. Finally, there is a need to more clearly define the organization's priorities both during and after crises, and to foster an understanding of the organization's minimum operational requirements (*Unity of purpose*) [22, 23]. A *Unity of purpose* seems particularly difficult to implement across borders in those times, as measures to overcome the crisis are often decided on and implemented at

the national level without sufficiently addressing cross-border needs [14, 15]. Cross-border governance could be a key enhancer of healthcare resilience in border regions [14].

Interestingly, we unveiled a relationship between organizational resilience and employees' mental health, suggesting that promoting the aforementioned resilience markers might not only improve healthcare institutions' performance (in times of crises) but also contribute to employee mental health. Prior research indeed linked organizational resilience to employee wellbeing [36]. A systematic review by Wei and colleagues linked a healthy work environment, reflected by organizational culture and patient care environments, to staff's psychological health, job satisfaction and retention as well as quality of patient care and patient safety [36]. In line, Montgomery et al. identified reduced mental health of healthcare staff as a key indicator for deficient hospital culture and inadequate organizational resources [37].

Regarding individual resilience, our results revealed that healthcare professionals exhibited normal resilience on average. However, 15.8% of participants reported low individual resilience and might benefit from interventions to promote this. Interventions promoting stress-coping, for instance, seem to have the potential to strengthen healthcare professionals' resilience and clinical performance: The introduction of stress management programs in 22 hospitals led to a significant reduction in medical errors and malpractice claims [38].

Interestingly, the percentage of healthcare professionals with low individual resilience (i.e., 15.8%) corresponds to those experiencing anxiety (22.1%) and depressive (15.9%) symptoms, implying a potential link. In line with previous research [39], we identified a significant correlation between low individual resilience and psychological distress, supporting the idea that resilience-promoting measures might preserve mental health.

To assess potential country-specific effects, we examined differences between Germany, Belgium and the Netherlands with respect to psychological distress, work-related stressors, and resilience. Healthcare professionals in Germany exhibited significantly higher rates of depression than in Belgium and reported significantly greater burden by work-related stressors than their colleagues in Belgium and the Netherlands. Similarly, German healthcare institutions demonstrated significantly lower organizational resilience than Belgian institutions in the domains *Situation awareness* and *Innovation and creativity*. These findings suggest differences in encouraging staff to monitor and report early warning signs (*Situation awareness*) as well as in applying creative problem-solving (*Innovation and creativity)* [22, 23]. The observed country differences in depression, stressors, and organizational resilience may be originated in (deficient) hospital culture and (inadequate) organizational resources [36, 37]. In addition, general cultural differences might contribute to country differences in organizational resilience, with Germany's stronger norms and lower tolerance for deviant behavior, possibly accounting for differences to Belgium in *Innovation and creativity* [40]. Furthermore, country-variations might also be influenced by differences in medical education. Compared to Germany, other European countries (e.g., the Netherlands, Belgium) have a higher percentage of academically trained nursing staff, allowing them to take greater responsibility in patient care [33], which may result in greater job satisfaction and appreciation.

With regard to the comparison of the different countries, it is noteworthy that there are some substantial differences in terms of stressors and resilience factors, although the pressures of cross-border care are presumably similar for all countries. Although the reasons for this cannot be conclusively clarified in our study, important conclusions for other European border regions can still be drawn from these results.

First, a comparison of stressors and resilience factors can be used to identify strengths and weaknesses in the respective national health systems. In addition to staff shortages, these may also include bureaucratic or organizational hurdles, for example.

Second, the results of such surveys can be used to organize cross-border support. Various approaches are conceivable here. For example, capacity bottlenecks can also be compensated for across borders. Furthermore, country-specific best-practice models could be identified in border regions and then transferred to other countries in the border region. Finally, synergies can be exploited by training organizational and individual resilience across borders (e.g. in Safety-II training courses).

As stated by Capello et al. [9], border regions frequently suffer from efficiency needs, i.e. they have a problem with the efficient use of their resources due to the border. In addition to legal and administrative obstacles, differences in organizational processes can also hinder the exploitation of existing potentials [41]. Our findings suggest that these may not affect all countries of a border region in an identical fashion, making a detailed analysis of the underlying reasons necessary.

In summary, a recommendation for other border regions may be to conduct comparable surveys of stressors and resilience factors and to systematically compare the countries with each other. The results can be used to create local risk profiles and develop specific solutions. Further investigation is needed for understanding the factors contributing to country-specific differences. The differences elucidated in our study underscore the potential of mutual learning across countries, facilitating the enhancement of resilience and the promotion of employee health.

### 4.1. Limitations

First, limited data from the Netherlands are available, due to a lower density of healthcare facilities. Second, to address concerns about survey length, an abbreviated survey was sent to Dutch healthcare facilities. Third, our study only examined individual and organizational resilience, lacking an assessment of team resilience. Future investigations of the dynamics between individuals, teams, and organizations are desirable. Future research might also focus on the comparison of needs between border regions and inland regions in healthcare to further delineate the specificity of border region's needs.

## 5. Conclusion

The findings highlight a significant number of healthcare professionals in the EMR facing mental health problems during the late phase of the COVID-19 pandemic. While healthcare professionals might benefit from programs promoting individual resilience, our results clearly show that main causes of stress arise from organizational issues. In order to nurture resilience in healthcare institutions in border regions such as the EMR, the management and provision of resources, the promotion of situation awareness and unified purposes during crises should be focused on in particular, while also considering country-specific needs. Our findings provide a foundation for resilience-promoting measures, highlighting the potential for transferring insights to other border regions.

## Supporting information

**S1 Data.**
(XLSX)

## Acknowledgments

We would like to express our gratitude to the participating healthcare professionals, without whose contribution this study would not have been realizable. Further, we would like to thank

the COMPAS consortium for their contribution to the project: Stefan Beckers[1,3], Juliët Beuken[5], Nadège Dubois[4], Kim Felsch[1,2], Alexandre Ghuysen[4], Ute Göretz[1,2], Zoé Kabanda[4], Katrin Kootz[1,3], Jule Kreitz[1,3], Andrea Lenes[1,2], Katharina Mohr[1,3], Fabio Pirkl[1,2], Cassandra Rehbock[1,3], Anna Riechenberg[1,2], Michelle Schmidt[1,2], Sebastian Sieberichs[1,2], Jolanda van Golde[5], Julia Varol[1,2], Daniëlle Verstegen[5], Corinna Wennmacher[1,3], Laura Wolff[1,2].

[1]Department of Anesthesiology, University Hospital RWTH Aachen, Medical Faculty, RWTH Aachen University, Aachen, Germany

[2]AIXTRA–Competence Center for Training and Patient Safety, Medical Faculty, RWTH Aachen University, Aachen, Germany

[3]ARS–Aachen Institute for Rescue Management and Public Safety, City of Aachen and University Hospital RWTH, Aachen, Germany

[4]Department of Public Health (DPH)–Center of Medical Simulation, University of Liège, Liège, Belgium

[5]School of Health Professions Education (SHE), Maastricht University, Maastricht, The Netherlands

## Author Contributions

**Conceptualization:** Leonie A. K. Loeffler, Sophie Isabelle Lambert, Lea Bouché, Saša Sopka, Lina Vogt.

**Data curation:** Leonie A. K. Loeffler, Martin Klasen.

**Formal analysis:** Leonie A. K. Loeffler.

**Funding acquisition:** Saša Sopka, Lina Vogt.

**Investigation:** Leonie A. K. Loeffler, Sophie Isabelle Lambert, Lea Bouché, Saša Sopka, Lina Vogt.

**Methodology:** Leonie A. K. Loeffler, Sophie Isabelle Lambert, Lea Bouché, Martin Klasen, Saša Sopka, Lina Vogt.

**Resources:** Saša Sopka, Lina Vogt.

**Software:** Leonie A. K. Loeffler, Sophie Isabelle Lambert.

**Supervision:** Saša Sopka, Lina Vogt.

**Visualization:** Leonie A. K. Loeffler.

**Writing – original draft:** Leonie A. K. Loeffler.

**Writing – review & editing:** Leonie A. K. Loeffler, Sophie Isabelle Lambert, Lea Bouché, Martin Klasen, Saša Sopka, Lina Vogt.

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
