## [Decision Letter · Decision Letter 0]

2 Aug 2024

PONE-D-24-21390Close to the border - Resilience in healthcare in a European border region: Findings of a needs analysisPLOS ONE

Dear Dr. Vogt,

Thank you for submitting your manuscript to PLOS ONE. After careful consideration, we feel that it has merit but does not fully meet PLOS ONE’s publication criteria as it currently stands. Therefore, we invite you to submit a revised version of the manuscript that addresses the points raised during the review process.

The reviewers have provided a number of comments for clarifying and improving the work, particularly with regard to the methodological and analytic decisions. Please pay particular attention to these comments in your revisions/response.

We look forward to receiving your revised manuscript.

Kind regards,

Jenny Wilkinson, PhD

Academic Editor

PLOS ONE

Journal Requirements:

Reviewers' comments:

Reviewer's Responses to Questions

**Comments to the Author**

1. Is the manuscript technically sound, and do the data support the conclusions?

Reviewer #1: Yes

Reviewer #2: No

2. Has the statistical analysis been performed appropriately and rigorously? 

Reviewer #1: Yes

Reviewer #2: N/A

3. Have the authors made all data underlying the findings in their manuscript fully available?

Reviewer #1: Yes

Reviewer #2: No

4. Is the manuscript presented in an intelligible fashion and written in standard English?

Reviewer #1: Yes

Reviewer #2: Yes

5. Review Comments to the Author

Reviewer #1: Referee report

Close to the border - Resilience in healthcare in a European border region: Findings of a needs analysis

This paper enters the vast literature on border effects from a novel (to me) angle, viz a quali-quantitative case study of one borer region with a specific focus on the needs of the healthcare industry.

As an economist I will provide comments that may be slightly different from what expected from the Author (s), hoping they will serve anyway the purpose of better clarifying the paper’s contribution.

1. For one thing, the need for this paper could be better explained by linking to the debate on the special needs of border regions. In Economics, this has gone through two main avenues:

a. Looking at whether regions on the border tend to trade disproportionately with other regions in the same Country, rather than with those at the same distance but beyond the border (see e.g. McCallum, 1995), or

b. Looking at whether regions located on the border tend to be less endowed with growth-enhancing factors (see e.g. Capello et al., 2018), or tend to exploit less efficiently similar endowments with factors (see e.g. Caragliu, 2022).

In both cases, I would suggest to link this to the extant debate.

2. The case for this study would also be strengthened by making it more explicit why healthcare industry employees should be more subject to, say, stress if located on the border. Presently, this is framed into a broader context whereby “Europe with its 47 countries is marked by numerous border regions and a relatively high population density, presenting unique challenges for healthcare institutions. In more detail, cross-border patient care and communication impose additional burdens on healthcare staff due to higher administrative costs arising through missing documents, language barriers, cultural disparities, and heightened demands on hospital infrastructure.” Yet, to the best of my knowledge, border-located hospitals do not serve patients from across the border, and when this does happen, it is typically the result of cooperation agreements between or among policy bodies from different country contexts (see e.g. Martínez, 2018). Again, why should healthcare employees be more stressed if working in a border hospital, rather than in the mainland of their home Countries?

3. Why focusing on the Meuse-Rhine border region? What are the advantages of this choice, and what, if any, its shortcomings?

4. Within this framework, policy implications could be strengthened by generalizing these findings and illustrating how other hospital managers in similar geographical and institutional locations may steer their communities to avoid the type of stress that the Meuse-Rhine case study illustrated.

References

Capello, R., Caragliu, A., & Fratesi, U. (2018). Measuring border effects in European cross-border regions. Regional Studies, 52(7), 986-996.

Caragliu, A. (2022). Better together: untapped potentials in Central Europe. Papers in Regional Science, 101(5), 1051-1086.

Martínez, E. (2018). “Medicine knows no borders”, retrieved online on Ju.8, 2024 at the URL https://interreg.eu/interreg-highlights/people/medicine-knows-no-borders/

McCallum, J. (1995). National borders matter: Canada-US regional trade patterns. The American economic review, 85(3), 615-623.

Reviewer #2: The design of the work is interesting and innovative even if it requires some revisions.

Describe and explain whether the tests could be administered online and how the online survey was carried out, times and methods.

6. PLOS authors have the option to publish the peer review history of their article (what does this mean?). If published, this will include your full peer review and any attached files.

Reviewer #1: No

Reviewer #2: No

---

## [Author Response · Author response to Decision Letter 0]

28 Nov 2024

Response to Reviewer 

Journal Requirements: 

Please ensure that your manuscript meets PLOS ONE's style requirements, including those for file naming. The PLOS ONE style templates can be found at  

https://journals.plos.org/plosone/s/file?id=wjVg/PLOSOne_formatting_sample_main_body.pdf and  

Response: We have customized the PLOS One’s style requirements to the best of our knowledge and belief in accordance with the specifications of the journal. 

Response: In consultation with all authors, we have followed your suggestion and uploaded the data files This means that everyone has agreed to a data sharing plan. 

Response: In accordance with reviewer’s 2 remark, we added the invitation e-mail of our online survey in four languages (German, French, Dutch and English) as Supporting Information files and included the respective caption. 

Reviewers' comments: 

Reviewer's Responses to Questions 

Comments to the Author 

1. Is the manuscript technically sound, and do the data support the conclusions? 

Reviewer #1: Yes 

Reviewer #2: No 

2. Has the statistical analysis been performed appropriately and rigorously? 

Reviewer #1: Yes 

Reviewer #2: N/A 

3. Have the authors made all data underlying the findings in their manuscript fully available? 

Reviewer #1: Yes 

Reviewer #2: No 

4. Is the manuscript presented in an intelligible fashion and written in standard English? 

Reviewer #1: Yes 

Reviewer #2: Yes 

5. Review Comments to the Author 

Reviewer #1: Referee report 

Close to the border - Resilience in healthcare in a European border region: Findings of a needs analysis 

This paper enters the vast literature on border effects from a novel (to me) angle, viz a quali-quantitative case study of one borer region with a specific focus on the needs of the healthcare industry. 

As an economist I will provide comments that may be slightly different from what expected from the Author (s), hoping they will serve anyway the purpose of better clarifying the paper’s contribution. 

1. For one thing, the need for this paper could be better explained by linking to the debate on the special needs of border regions. In Economics, this has gone through two main avenues: 

a. Looking at whether regions on the border tend to trade disproportionately with other regions in the same Country, rather than with those at the same distance but beyond the border (see e.g. McCallum, 1995), or 

b. Looking at whether regions located on the border tend to be less endowed with growth-enhancing factors (see e.g. Capello et al., 2018), or tend to exploit less efficiently similar endowments with factors (see e.g. Caragliu, 2022). 

In both cases, I would suggest to link this to the extant debate. 

Response: We thank the reviewer for the valuable suggestion, and we welcome the input from the field of economics very much. We believe that linking the debate to the special needs of cross-border regions the paper’s added value becomes clearer. Looking beyond the horizon greatly enhances our manuscript. Therefore, we added to the manuscript the following in the tracked- changes mode: 

Page 5 lines 69-78 

“In economics, the special needs and characteristics of border regions have been subject. This debate has two approaches: firstly borders lead to regions trading disproportionately with other regions in the same country and not with those that are the same distance away but on the other side of the border [8]. Secondly border regions tend to be less endowed with growth-promoting factors [9] or utilize similar endowments less efficiently [9]. Economic growth, in turn, appears to be linked to improvements in the healthcare system, for example due to better public healthcare infrastructure [10]. Hence, research about patient care in border regions is important, to identify factors for improvements to enhance efficiency and quality of patient care. 

Looking back at the COVID 19 pandemic border regions are” 

2. The case for this study would also be strengthened by making it more explicit why healthcare industry employees should be more subject to, say, stress if located on the border. Presently, this is framed into a broader context whereby “Europe with its 47 countries is marked by numerous border regions and a relatively high population density, presenting unique challenges for healthcare institutions. In more detail, cross-border patient care and communication impose additional burdens on healthcare staff due to higher administrative costs arising through missing documents, language barriers, cultural disparities, and heightened demands on hospital infrastructure.” Yet, to the best of my knowledge, border-located hospitals do not serve patients from across the border, and when this does happen, it is typically the result of cooperation agreements between or among policy bodies from different country contexts (see e.g. Martínez, 2018). Again, why should healthcare employees be more stressed if working in a border hospital, rather than in the mainland of their home Countries? 

Response: We thank the reviewer for this important remark, and the reference to strengthen the manuscript. In the Euregio Meuse-Rhine, patients receive medical care across borders. 

Under EU Directive 2011/24/EU, healthcare systems within EU countries are encouraged to facilitate cross-border healthcare. This directive not only enables patients to seek safe and high-quality medical care in neighboring EU countries but also allows them to receive reimbursement for these services through their own national health insurance. Additionally, the directive actively promotes collaboration between national healthcare systems, thereby creating frameworks for cross-border-regional cooperation in healthcare as seen in our region. As experienced clinicians, some of our co-authoring colleagues can report from their own experience of working as emergency physicians in our region: They have been involved in several medical emergencies in e.g. the Netherlands and have transported the patient to the maximum care facility on the other side of the border for further treatment. This possibility of cross-border healthcare in our region is officially regulated via several agreements between governmental and healthcare organizations within the Euregio Meuse-Rhine and the EMRIC consortium. 

We argue that these unique conditions in cross-border-healthcare do place additional demands on healthcare professionals in border regions in terms of administrative, linguistic and logistical demands, which distinguishes their work environment from that of healthcare professionals in non-border regions. Thank you very much for pointing our attention to the interesting article by Martínez et all, 2018. 

We therefore added to the manuscript the following: 

Page 5 lines 90-92 

“Europe with its 47 countries is marked by 45 land border regions and 17 maritime border regions, meaning 62 overall border regions. Within this region a relatively high population density exists, presenting unique challenges for healthcare institutions.” 

3. Why focusing on the Meuse-Rhine border region? What are the advantages of this choice, and what, if any, its shortcomings? 

Response: Thank you for this important remark. 

We argue that the Meuse-Rhine border region offers a unique setting for studying resilience among healthcare professionals due to its cross-border healthcare systems, policies, and cultural and linguistic differences. These differences allow for an understanding of how healthcare professionals in cross-border regions adapt to challenges, with insights into which national policies may strengthen or hinder resilience. 

Furthermore, the Meuse-Rhine region’s tradition of cross-border collaboration offers insights on how (shared) resources and practices may influence resilience, making it relevant for other countries or cross-border regions with multicultural or multidisciplinary teams facing similar issues like language barriers and administrative and cultural differences. 

However, while many of our findings on psychological well-being and the effects of workload pressures in healthcare can inform global best practices in building psychological and organizational resilience, the specific socio-political and economic contexts of Belgium, the Netherlands, and Germany may limit generalizability, especially to low- and middle-income countries. Nonetheless, the core principles of resilience-building remain adaptable to diverse healthcare systems worldwide. 

The publication by Robert Capello et all. 2018 shows efficiency needs and endowment needs in their work. Figures 1 a and b of the appendix clearly show the high needs (cultural events efficiency below border region) and the equally high endowment needs (cultural events below border region average.) These data show that the Euregio Meuse-Rhine requires special attention and that this region, like several others, is ideally suited for such studies. 

In addition, the three countries of the Euregio Maas- Rhine are very similar in cultural, political and economic terms (all EU countries, Schengen area, etc.) and there is a lively exchange (tourism, commuters...) confirmed by McCullum. Whether this is an advantage or disadvantage of the study depends on one's perspective. On the one hand, the region is presumably representative of comparable EU border regions, but on the other hand it may not be representative of border regions elsewhere in the world. 

One of the advantages is the proximity of the team of authors to this region. They have been working, living and researching here for years and can therefore contribute first-hand insights.

---

## [Decision Letter · Decision Letter 1]

6 Dec 2024

Close to the border - Resilience in healthcare in a European border region: Findings of a needs analysis

PONE-D-24-21390R1

Dear Dr. Vogt,

We’re pleased to inform you that your manuscript has been judged scientifically suitable for publication and will be formally accepted for publication once it meets all outstanding technical requirements.

Kind regards,

Jenny Wilkinson, PhD

Academic Editor

PLOS ONE

Additional Editor Comments (optional):

Reviewers' comments:

Reviewer's Responses to Questions

**Comments to the Author**

1. If the authors have adequately addressed your comments raised in a previous round of review and you feel that this manuscript is now acceptable for publication, you may indicate that here to bypass the “Comments to the Author” section, enter your conflict of interest statement in the “Confidential to Editor” section, and submit your "Accept" recommendation.

Reviewer #1: All comments have been addressed

2. Is the manuscript technically sound, and do the data support the conclusions?

Reviewer #1: Yes

3. Has the statistical analysis been performed appropriately and rigorously? 

Reviewer #1: Yes

4. Have the authors made all data underlying the findings in their manuscript fully available?

Reviewer #1: Yes

5. Is the manuscript presented in an intelligible fashion and written in standard English?

Reviewer #1: Yes

6. Review Comments to the Author

Reviewer #1: Dear Authors,

Thank you for addressing all my initial concerns..the paper can now be accepted.

Good luck with the impact of your work!

7. PLOS authors have the option to publish the peer review history of their article (what does this mean?). If published, this will include your full peer review and any attached files.

Reviewer #1: No

---

## [Editor Report · Acceptance letter]

8 Jan 2025

PONE-D-24-21390R1 

PLOS ONE

Dear Dr. Vogt, 

I'm pleased to inform you that your manuscript has been deemed suitable for publication in PLOS ONE. Congratulations! Your manuscript is now being handed over to our production team.

Kind regards, 

on behalf of

Dr Jenny Wilkinson 

Academic Editor

PLOS ONE